# Molecular-scale dynamics of light-induced spin cross-over in a two-dimensional layer

Kaushik Bairagi[1], Olga Iasco[2], Amandine Bellec[1], Alexey Kartsev[3,4], Dongzhe Li[3], Jérôme Lagoute[1], Cyril Chacon[1], Yann Girard[1], Sylvie Rousset[1], Frédéric Miserque[5], Yannick J. Dappe[3], Alexander Smogunov[3], Cyrille Barreteau[3], Marie-Laure Boillot[2], Talal Mallah[2] & Vincent Repain[1]

Spin cross-over molecules show the unique ability to switch between two spin states when submitted to external stimuli such as temperature, light or voltage. If controlled at the molecular scale, such switches would be of great interest for the development of genuine molecular devices in spintronics, sensing and for nanomechanics. Unfortunately, up to now, little is known on the behaviour of spin cross-over molecules organized in two dimensions and their ability to show cooperative transformation. Here we demonstrate that a combination of scanning tunnelling microscopy measurements and *ab initio* calculations allows discriminating unambiguously between both states by local vibrational spectroscopy. We also show that a single layer of spin cross-over molecules in contact with a metallic surface displays light-induced collective processes between two ordered mixed spin-state phases with two distinct timescale dynamics. These results open a way to molecular scale control of two-dimensional spin cross-over layers.

[1] Laboratoire Matériaux et Phénomènes Quantiques, Univ Paris Diderot, Sorbonne Paris Cité, CNRS, UMR 7162, 75013 Paris, France. [2] Institut de Chimie Moléculaire et des Matériaux d'Orsay, Univ Paris Sud, Univ Paris Saclay, CNRS, UMR 8182, 91405 Orsay Cedex, France. [3] Service de Physique de l'Etat Condensé, DSM/IRAMIS/SPEC (CNRS UMR 3680), CEA Saclay, Univ Paris Saclay, 91191 Gif sur Yvette Cedex, France. [4] National Research Tomsk State University, 36, Lenina pr., Tomsk 634050, Russia. [5] CEA/DEN/DANS/DPC/SCCME, Laboratoire d'Etude de la Corrosion Aqueuse, F-91191 Gif-sur-Yvette, France. Correspondence and requests for materials should be addressed to A.B. (email: amandine.bellec@univ-paris-diderot.fr).

Spin cross-over (SCO) molecules are promising systems for the development of molecular spintronics, as they present two electronic spin states that can be controlled by external stimuli such as light, pressure, temperature or electric field[1–3]. The behaviour of $Fe^{II}$ SCO molecules in crystalline solids is well documented, especially the low-spin (LS, spin $S = 0$) to high-spin (HS, spin $S = 2$) state conversion induced at low temperature by the light-induced excited spin-state trapping (LIESST) effect[4] or by thermal process. The structural distortions that accompany the spin-state switching between $(t_{2g})^6(e_g)^0$ and $(t_{2g})^4(e_g)^2$ configurations produce elastic strains manifested by long-range propagative effects and cooperative transformations. For this reason, the incorporation of SCO molecules in electronic devices[5] requires a deep understanding of their properties from the sub-micrometre and nanometre scale[2,6] down to the molecular scale[7–9].

Recent investigations of ultra-thin layers on substrates indicate strong modifications of SCO molecule properties with the possible coexistence of the two spin states[9–14] and even the suppression of the spin state switching[10]. Furthermore, light irradiation of single crystals produces a transformation process with macroscopic interface movements, whose underlying basic mechanism has been established[15–17]. The surrounding medium of SCO molecules being active, one challenging issue to investigate is the photo-induced transformation process at the level of the molecular monolayer[18,12]. Although X-ray absorption techniques can provide valuable information, they remain macroscopic and present important drawbacks such as the degradation of molecules and soft X-ray induced spin-state trapping. Therefore, scanning tunnelling microscopy (STM) stands as a unique technique to access the direct visualization of photo-induced spin transition at the molecular scale. From theoretical point of view, SCO molecules have been studied in several works within the so-called DFT $+ U$ approach, which combines the density functional theory (DFT) with the Hubbard $U$ onsite term (applied on localized orbitals of magnetic atom), necessary to correct for self-interaction errors. For example, Lebegue et al.[19] and Paulsen[20] have used GGA $+ U$ to study molecular crystals of Fe-based SCO molecules. In addition, DFT studies of SCO molecules on metallic substrates[21] or bidimensional materials[22] have been recently presented. As a result, the importance of the $U$ parameter for describing properly the stability and LS to HS transition was pointed out.

Here we focus our study on $[Fe^{II}((3, 5-(CH_3)_2Pz)_3BH)_2]$ (**1**)(Pz = pyrazolyl) sub-monolayers grown on Au(111) substrate and their characterization by STM and scanning tunnelling spectroscopy (STS) at 4.6 K. Ab-initio calculations allow to unambiguously assign the spin state of the molecules on the surface, as well-pronounced inelastic steps in low-energy spectra are predicted only for molecules in HS state, in agreement with the experimental STS spectra. This local inelastic spectroscopy should provide a more versatile way of determining the spin state of spin cross-over molecules than the zero bias Kondo feature observed so far, which required a strong coupling with a metallic substrate and low-temperature measurements. We thus provide evidences of the formation of a long-range order superstructure alternating the presence of one molecule in HS state and two molecules in the LS state, which is an original thermodynamic phase of those spin cross-over molecules in low dimension as compared with the bulk. We show that light illumination of such molecular islands induces the LS to HS switching of the molecules at the surface scale. STM measurements enable us to access the molecular scale dynamics of the spin-state switching, thus showing the importance of propagative effects and the internal fluctuations of the excited phase at 4.6 K, which cannot be extracted from other global surface techniques such as X-ray absorption spectroscopy[12,18].

## Results

**HS and LS molecular assembly at low temperature.** In bulk, SCO molecule **1** (Fig. 1a) present a transition from LS to HS at a temperature of ca. 186 K or by LIESST effect[23]. Our theoretical DFT $+ U$ study confirms that the molecular magnetism is controlled by the Fe-N distance—the longer it is, the more favourable is the HS solution (see Supplementary Fig. 1)—a well-known fact resulting from the competition between the Hund's rule coupling and the Fe $d$-levels splitting in a crystal field. The inclusion of $U$ on the Fe $d$-orbitals does not change significantly HS and LS atomic configurations but reduces significantly the HS–LS energy separation, from 1.2 eV ($U = 0$ eV) to 0.6 eV ($U = 2$ eV) (see Supplementary Fig. 1), favouring further the HS state.

Deposited on a Au(111) substrate with a sub-monolayer coverage, the molecules **1** form self-assembled islands of one monolayer height (see Supplementary Fig. 2) with a spin-dependent superstructure that can be observed for specific scanning voltages. At $-1.5$ V (Fig. 1b, inset), all the molecules of the lattice are visible. Both chiralities for the molecular network are observed in equal proportions on the surface, that is, with a direct or an indirect angle between the lattice vectors **A** and **B**. The two-dimensional lattice is characterized by $\|A\| = 8.6 \pm 0.2$ Å, $\|B\| = 10.6 \pm 0.3$ Å, $\|(A, B)\| = 80 \pm 2°$. Those parameters have been obtained by a Fourier transform analysis of large-scale STM images with molecular resolution, recorded on 18 different molecular islands. The given values are the average and the error bars are the standard deviations. It is worth noting that the monolayer unit cell parameters are close to those of the crystallographic plane (0 1 $\bar{1}$) of the bulk structure at 298 K[24], where the molecules have their pseudo $C_3$ axis almost lying within the plane. This particular molecular orientation has an impact on the shape of the STM images as discussed below. No commensurability between the molecular lattice and the Au(111) substrate is observed. At 0.3 V, in both scanning modes, a superstructure ($S_{1/3}$) is observed with only one molecule over three appearing bright (Fig. 1b and Supplementary Fig. 3 for details). Both images (Fig. 1b and inset) have the same orientation and scale, thus allowing a direct comparison between the molecular network and its superstructure. The $S_{1/3}$ superstructure is a $\begin{pmatrix} 2 & -1 \\ 1 & 1 \end{pmatrix}$ reconstruction containing three molecules per unit cell with measured lattice parameters $\|a\| = 18.7 \pm 0.3$ Å, $\|b\| = 14.6 \pm 0.4$ Å, $\|(a, b)\| = 78 \pm 3°$. Thus, three energetically equivalent phases can be observed for the $S_{1/3}$ superstructure. The superstructure signature observed by STM at 0.3 V clearly arises from electronic effects and can be related to the $Fe^{II}$ spin state as demonstrated in the following.

**Electronic contrast and inelastic tunnelling spectroscopy.** In the case where the molecules have a relatively large electronic interaction with the substrate, it is possible to discriminate between both molecular spin states, as HS molecules would present a Kondo anomaly at zero bias in low-energy STS spectra[9,13]. When the electronic interaction between the substrate and the molecules is weak, the Kondo anomaly can disappear at 4.6 K and the assignment of HS and the LS states remains an open question. To assign the spin state of the bright and dark molecules at 0.3 V and understand their round shape observed by STM, we performed ab initio density functional calculations following a well-established DFT $+ U$ procedure. As already discussed above, the magnetism is favoured by $U$, but the main physical parameter underlying the magnetic transition is the Fe–N distance. As the exact value of $U$ remains unknown (usually it is in the range between 1.5 and 3 eV for Fe), we will present and discuss in the

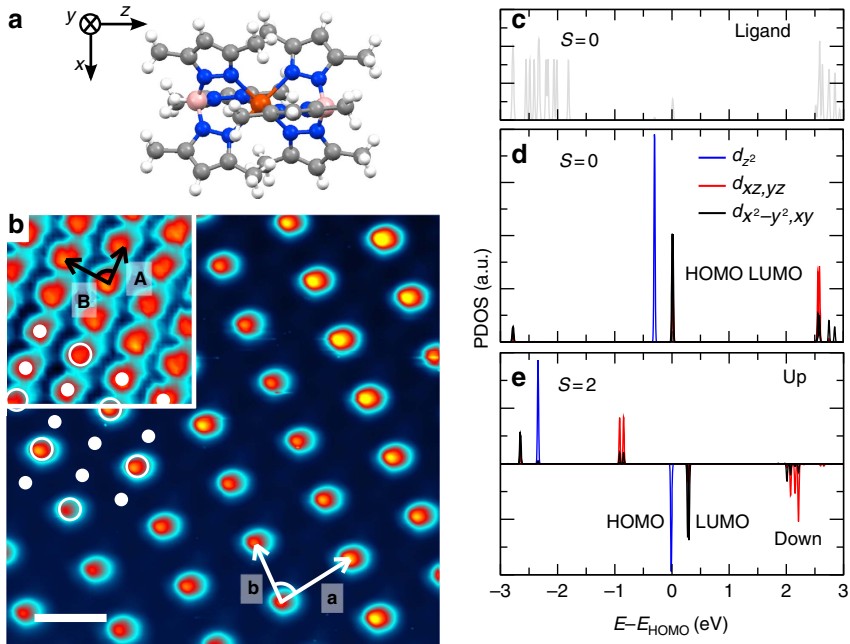

**Figure 1 | Ordered phase of HS and LS molecules. (a)** Schematic of molecule **1** with the $z$ axis along the B–Fe–B direction (grey: C atoms, blue: N atoms, pink: B atoms, red: Fe atom and white: H atoms). **(b)** STM image acquired at 0.3 V in constant height mode showing the mixed spin-state superstructure $S_{1/3}$ ($<I> = 50$ pA). Inset, topographic STM image acquired at $V = -1.5$ V showing the full molecular crystal ($I = 20$ pA). The scale bar corresponds to 2 nm and is common for both images. The lattice vectors of the molecular network (**A** and **B**, black) and the $S_{1/3}$ superstructure (**a** and **b**, white) are indicated. The full (empty) dots indicate the position of the dark (bright) molecules at 0.3 V. **(c)** Projected density of states (PDOS) on the ligand atoms for the LS state. **(d)** PDOS on the $d$-orbitals of the central Fe$^{II}$ atom for **1** in LS ($S = 0$). **(e)** PDOS on the $d$-orbitals of the central Fe$^{II}$ atom for **1** in HS ($S = 2$) states. For **c**–**e**, the energy of HOMOs is set as zero.

following the results of DFT calculation without $U$, to get physical interpretation of experiments. However, we have checked that the major influence of $U$ on the electronic structure is an increase of a highest occupied molecular orbital (HOMO)–lowest unoccupied molecular orbital (LUMO) gap, whereas the main physics at the orbital level remains unchanged (see Supplementary Fig. 4). We present in Fig. 1c–e the density of states projected onto the ligand states and the different Fe $d$-orbitals for both spin states at equilibrium (see Supplementary Fig. 5 for the separated curves). These results clearly show that, depending on the molecular spin, the HOMO and LUMO orbitals are related to different $d$-orbitals. In particular, the two-fold degenerate $d_{x^2-y^2}$, $d_{xy}$ states mainly correspond to the HOMO in the LS state and to the LUMO in the HS state. The $d$-states of this orientation are of importance, as for the deposited molecule with its $z$ axis (B–Fe–B axis) along the Au surface in agreement with the (01$\bar{1}$) bulk plane they would point towards the STM tip. This important point is confirmed for the LS molecule deposited on the Au(111) surface (see Supplementary Fig. 4), where only $d_{x^2-y^2,x,y}$ orbitals are found to be seen in the vacuum above the molecule, while no signal is observed for the $d_{z^2}$-originated states (we did not calculate the deposited molecule in the HS state, as it needs a locally constraint magnetic calculation, which is not yet implemented in our code; otherwise, it converges to the lowest energy LS state). It has been also found that molecular levels and their relative positions are only weakly modified by adsorption on the Au surface (the adsorption energy was found to be ~2.1 eV, which also indicates a rather weak molecule–substrate interaction), justifying thus our free-molecule analysis. Hence, the projected density of states (Fig. 1d,e) indicates that at positive voltage (below 1 V) the tunnelling conductance is expected to be higher for the HS molecules (see Supplementary Fig. 3a for measured differential conductance). As a consequence, the bright spots in the STM

images at 0.3 V can be ascribed to the molecules in the HS state. In contrast, images at negative bias show the molecules in the LS state (see Supplementary Fig. 3c). Moreover, the $d_{x^2-y^2}$, $d_{xy}$-derived orbitals are more localized on the central Fe atom and will be rather featureless far away from the molecule (see Supplementary Fig. 5), which accounts for the round molecular shapes observed by STM, the tunnelling conduction being possible at that energy through the Fe atom and not through the ligand[9,10,13].

Figure 2a presents the low-energy spectra recorded on both types of molecules in the $S_{1/3}$ superstructure. For the molecules appearing bright at 0.3 V, which are in HS state, two inelastic steps are observed at $18 \pm 2$ and $67 \pm 2$ meV (marked as 1 and 2 in Fig. 2a, respectively). For the dark molecules ascribed to LS state, only a small signature at $67 \pm 2$ meV is observed. It is worth noting that, contrary to previous works[9,13], no Kondo feature is clearly observed, as the central Fe$^{II}$ atom is most probably decoupled from the Au(111) surface.

To understand the inelastic steps observed in the low-energy spectra, we calculated the electron–vibration couplings for the free molecule in both spin states. Figure 2b shows the coupling constants for vibrational modes associated to the Fe $d_{x^2-y^2,xy}$-derived molecular orbitals, which contribute strongly to the electron transport at low voltage as argued above. In the energy range below 100 meV, only two vibrational modes at ~12 and 42 meV with relatively high electron–vibron coupling in the case of HS state and no well-coupled modes for the LS state are resolved (Fig. 2b). The corresponding vibration modes of the free-standing HS molecule mostly have longitudinal in-plane polarization vectors (directed along the B–Fe–B axis) and therefore should not be suppressed by the substrate (Fig. 2c). Those active vibronic modes will affect the transport properties of the deposited HS molecules by the electron–vibration scattering process that leads to the presence of the inelastic steps in the

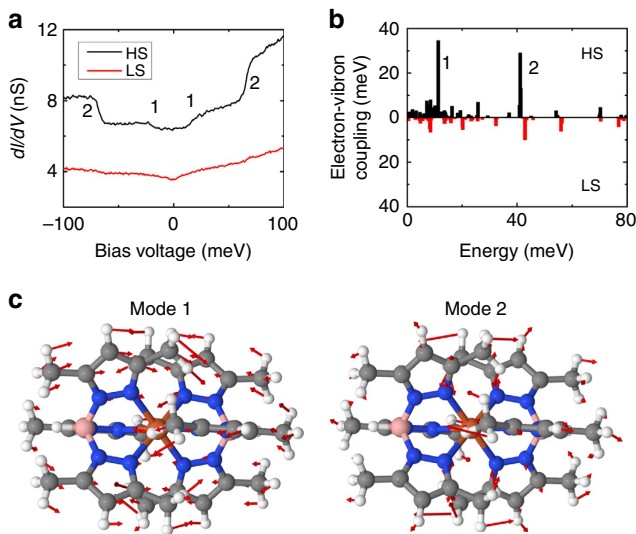

**Figure 2 | Molecular vibrational modes for both spin states.**
(**a**) Low-energy $dI/dV$ curves for both types of molecules in the $S_{1/3}$ superstructure. 1 and 2 mark the position of the pronounced inelastic steps observed for the molecules in the HS state. (**b**) DFT coupling constants of the vibrational modes associated with the $d_{x^2-y^2,xy}$-derived molecular orbitals calculated for the free molecule in both spin states. (**c**) The displacement vectors (red arrows) for the two active modes indicating their longitudinal character (atoms move mostly along the B–Fe–B axis).

$dI/dV$ spectra. It is noteworthy that the frequencies of these modes could be slightly altered by the physisorption on the substrate, which may explain the difference in energy position between the experimental curves and the calculated modes. Thus, the assignment of the HS (LS) states to the bright (dark) molecules seen at $V = 0.3$ V is reinforced by low-energy spectroscopy and the insight drawn from theoretical analysis of electron–vibration couplings.

In the literature, such mixed spin-state phases at low temperature were reported for spin cross-over materials in bulk[25,26]. The co-existence of the both SCO molecule spin states has already been observed once deposited on surfaces[9–14]. However, remarkably, here the molecules of different spin states self-organize in a long-range ordered phase at low temperature with domain size up to $200 \times 200$ nm$^2$.

**Light-induced excited spin-state trapping at molecular scale.**
We characterize the response of the molecule **1** islands to blue light illumination (405 nm). This energy corresponds to the low-energy side of the Fe$^{II}$ to ligand charge-transfer absorption band[27,28] that triggers the LS to HS transition by LIESST. Thus, although scanning the same area at a bias voltage of 0.3 V with an acquisition time of 5 min and 51 s per image, we shine the sample with an external blue diode for 10 h and then we let it relax for another 10 h. It is noteworthy that during the experiment the thermal drift in the STM images is small enough to follow the same area. The experiment is summarized in Fig. 3 and the movie is available as a Supplementary Movie 1. Initially, the scanned area is covered by the almost defect-free $S_{1/3}$ superstructure as visible in Fig. 3a. The blue light clearly induces the formation of another superstructure ($S_{1/2}$) with one bright molecule and one dark molecule per unit cell as can be seen in Fig. 3b after 9 h and 45 min under illumination. By comparing with the LIESST effect in bulk, the blue light inducing the transition from the dark molecules into bright ones confirms that the molecules appearing bright at 0.3 V can unambiguously

be ascribed to molecules in HS state. This also proves the possibility of observing a single layer of SCO in direct contact with a metallic substrate. After 10 h, the blue illumination is stopped and the system relaxes to its initial state, that is, the $S_{1/2}$ superstructure disappears to the profit of the $S_{1/3}$ one (Fig. 3c). To exclude any influence of the scanning conditions, we image a $S_{1/3}$ area for 14 h at 0.3 V without any external illumination. During all experiments the $S_{1/3}$ superstructure is kept intact, thus proving that the used scanning conditions are non-invasive. By shining a red light (632.8 nm) on the $S_{1/3}$ area, the superstructure is also preserved, which is consistent with the very weak absorption cross-section characterizing molecule **1** at this energy ($^1A_1 \rightarrow {}^1T_1$ ligand-field absorption expected at *ca.* 530 nm)[27,28].

The $S_{1/2}$ photoinduced superstructure is a $\begin{pmatrix} 1 & -1 \\ 1 & 1 \end{pmatrix}$ reconstruction of the molecular layer with a unit cell composed of one molecule in HS state and one molecule in LS state, and with lattice parameters $\|\mathbf{c}\| = 12.2 \pm 0.3$ Å, $\|\mathbf{d}\| = 14.5 \pm 0.3$ Å and $\|(\mathbf{c}, \mathbf{d})\| = 99 \pm 3°$. Both superstructures are schematized in Fig. 3g on top of the full crystal network, $S_{1/3}$ in red and $S_{1/2}$ in blue.

Qualitatively, one can see in the STM images (Fig. 3b) that the $S_{1/2}$ superstructure forms small domains—inferior to $10 \times 10$ nm$^2$—in contrast to the $S_{1/3}$ superstructure, and even under blue light we never observed an area with all the molecules in HS state. Stopping the blue illumination induces the relaxation of the $S_{1/2}$ superstructure into the stable $S_{1/3}$ one. As seen in Fig. 3c acquired 9 h and 45 min after stopping the blue light, large $S_{1/3}$ domains are observed with different phases and they are separated by small $S_{1/2}$ domains. As the relaxation of the $S_{1/2}$ superstructure implies the formation of phase-shifted $S_{1/3}$ domains, the defect-free $S_{1/3}$ superstructure will only be fully recovered after the complete shrinking of the smaller $S_{1/3}$ domains to the profit of the larger ones. Annealing at room temperature induces the complete recovery of the $S_{1/3}$ superstructure over large distances.

To analyse more quantitatively the kinetics of the SCO transition, we follow the intensity of the characteristic peaks of both superstructures in the Fourier space. Figure 3d–f present the Fourier-transformed images corresponding to the STM images in Fig. 3a–c, respectively. In Fig. 3h, the reciprocal lattices of $S_{1/3}$ and $S_{1/2}$ are schematized in red and blue, respectively. We follow in time the intensity of the (1,0) and (0,1) peaks of each superstructure. As expected, the intensity of the $S_{1/3}$ peaks decreases with time under blue illumination, whereas the $S_{1/2}$ peak intensity increases. For the four peaks, we define as in ref. 29 a dimensionless parameter $x(t) = \frac{I(t) - I(t=0)}{I(t=\infty) - I(t=0)}$. The average curve of the four normalized peaks is presented in Fig. 3i (black squares). Using a monoexponential fit, we access the characteristic time of $114 \pm 8$ min. The elastic interactions between SCO molecules most probably play an important role in the $S_{1/3}$ to $S_{1/2}$ transition. In the movie, the propagation of the $S_{1/2}$ domains seems to develop from one corner of the images to the other. Such 'transition front' can be observed in SCO solids[17]. For its part, the thermal relaxation is fitted in the Hauser model[4] by a monoexponential decay, thus giving a characteristic time of $131 \pm 5$ min. This value can be compared with the photoinduced HS state mean lifetime of a few minutes for **1** in bulk, which is extrapolated from the correlation between the low-temperature tunnelling rate constant and the transition temperature (*ca.* 186 K for **1**) established by Hauser[4].

If the characteristic excitation and relaxation times are clearly visible in the experimental data, fluctuations of $x(t)$ are also observed around the final state. The illumination tends to favour

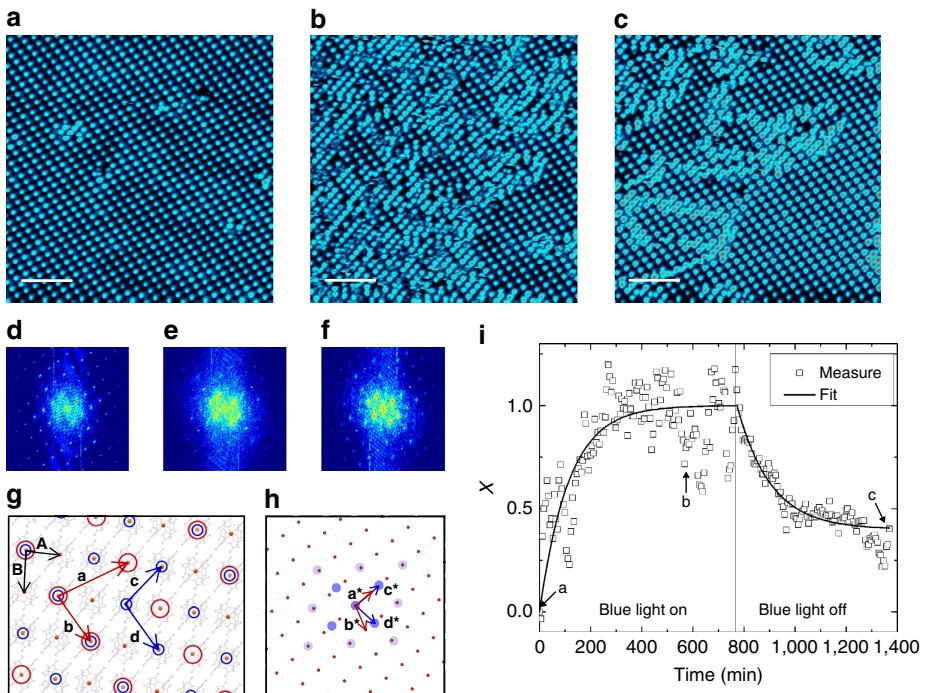

**Figure 3 | Light-induced SCO and thermal relaxation at the molecular scale.** (**a**–**c**) STM images (4.6 K) of the same area (**a**) in its initial state presenting the $S_{1/3}$ superstructure, (**b**) under blue light illumination after 9 h and 45 min of exposure and (**c**) in its relaxed state 9 h and 45 min after stopping the blue light illumination ($V = 0.3$ V, $I = 20$ pA). Scale bars, 10 nm. (**d**–**f**) Fourier-transformed images of **a**–**c**, respectively. (**g**) Model of the molecular network and both superstructures. For the full crystal representation, the Fe$^{II}$ centres are enlightened and the lattice vectors **A** and **B** are indicated in black (indirect **A** and **B** angle), the $S_{1/3}$ superstructure is represented by the red circles and lattice vectors **a** and **b** the $S_{1/2}$ one by the blue circles and lattice vectors **c** and **d**. (**h**) Reciprocal space of both $S_{1/3}$ and $S_{1/2}$ superstructures (same colour code). (**i**) Time evolution of the normalized peak intensities $x$. The data (squares) are fitted using a least squares method by mono-exponential increase and decrease under blue light and for the thermal relaxation, respectively (solid lines). The arrows indicate the position of the STM images (**a**–**c**).

the LS to HS conversion of the molecules, whereas, as evidenced by the small size of the $S_{1/2}$ domains, the $S_{1/3}$ superstructure is more stable than the $S_{1/2}$ one. This leads to an intrinsic dynamics of the system visible in the movie. To quantify this internal dynamics we follow the apparent height of molecules with time, that is, over the consecutive STM images. Technically, this has been done by subtracting a mean base plane to all raw STM images and measuring the mean height over a small region of interest that stays focused on the very same molecule for more than hundred images. Figure 4a presents an STM image taken under blue illumination, in which we select different molecules. Molecules A (in HS state) and B (in LS state) are located in an $S_{1/3}$ area preserved under illumination, that is, no change of the spin state is observed (zoom in Fig. 4b). Molecule C, for its part, is located in the $S_{1/2}$ superstructure (zoom in Fig. 4c).

As visible in Fig. 4d, the apparent height corresponding to a combination of topographic and electronic effects of molecules A and B is followed over time with and without illumination. When the illumination is off, the apparent height is constant but the noise increases as soon as the illumination is turned on. Even though molecules A and B keep their spin state, they fluctuate due to the propagation of the elastic constraints induced by the formation of the $S_{1/2}$ domains. This noise even increases as the $S_{1/2}$ domains are developing and propagating towards molecules A and B.

Under illumination, the apparent height of molecule C located in a $S_{1/2}$ area (Fig. 4e) varies between two values corresponding to the molecule in HS state (around 1 Å) and the LS state (around −1 Å) (the variations are measured around the mean plane of each images). From the observed telegraphic noise, two well-defined durations $t_{LS}$ and $t_{HS}$ can be defined (see Fig. 4e).

They represent respectively the durations that the molecules stay in LS (HS) state before switching to HS (LS) state. The distribution of $t_{LS}$ and $t_{HS}$ shown in Fig. 4f,g are measured over more than 100 molecules in $S_{1/2}$ domains. This method enables us to have a statistics over a large number of molecules even though the time resolution is limited to the image acquisition time (5 min and 51 s per image). Both distributions follow an exponential decay with similar characteristic times of $10 \pm 1$ min. Interestingly, we observe simultaneous and opposite switching of the molecules in adjacent sites of the $S_{1/2}$ superstructure, which underlines short-range correlations. These relatively fast fluctuations (10 min) compared with the transition of the domain structures from $S_{1/3}$ to $S_{1/2}$ (114 min) reflect the internal dynamics of the system and the metastability of the $S_{1/2}$ superstructure. Indeed, as the volume of the HS molecules is larger than the volume of the LS ones, the LS to HS switching induces local distortions and related strains, which prevent both the complete conversion of LS molecules to HS ones in the two-dimensional crystal and the $S_{1/2}$ superstructure to develop over large area even under continuous illumination. The intrinsic dynamics of the $S_{1/2}$ superstructure is most probably due to the competition between the LS to HS transition and the development of the stable $S_{1/3}$ superstructure.

## Discussion

We show that by combining theoretical calculations with STM and STS measurements, it is possible to assign unambiguously the spin state of molecules even in the absence of Kondo anomaly, which constitutes a general way to proceed. We thus demonstrate

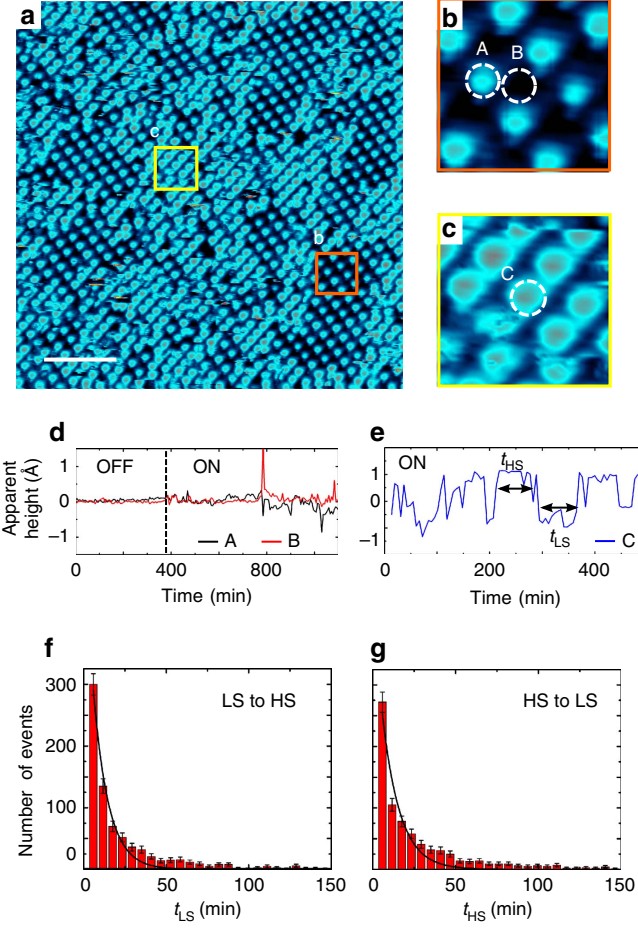

**Figure 4 | Internal dynamics of the photoexcited phase.** (**a**) STM image taken under blue illumination after 11 h and 15 min of exposure ($V = 0.3$ V, $I = 20$ pA). Scale bar, 10 nm. (**b**) Zoom on a $S_{1/3}$ area, marked by a orange square in **a**. (**c**) Zoom on a $S_{1/2}$ area, marked by a yellow square in **a**. (**d**) Apparent heights of molecules A and B, displayed in **b** versus time. Their apparent heights are followed with the illumination off and on. (**e**) Apparent height of molecule C, displayed in **c**, which switches between LS and HS states under continuous illumination. (**f,g**) Distribution of $t_{LS}$ and $t_{HS}$, respectively. $t_{LS}$ ($t_{HS}$) is the duration for a molecule in a LS (HS) state before switching in a HS (LS) state. The size of the boxes corresponds to the acquisition time of a STM image (5 min 51 s).

the formation of a long-range ordered mixed spin-state phase. In addition, we observe, for the first time at the molecular scale, light-induced SCO with the propagation of the excited phase, which is the signature of collective effects at the level of a single layer of SCO molecules. We also characterize the internal dynamics of the excited system and its thermal relaxation. These findings open the routes for the control of bistability of individual SCO molecules organized on a metallic substrate, which is one of the challenges in this area of research.

## Methods

**Chemical synthesis.** $[Fe^{II}((3, 5-(CH_3)_2Pz)_3BH)_2]$ (**1**) powder was synthesized using an improved synthesis described by Davesne et al.[30]. The precipitate was washed with methanol, dried under vacuum and recrystallized from tetrahydrofuran before any sublimation, to give colourless crystals of **1**. The element analysis, mass spectrum and powder X-ray diffractogram are satisfying.

*Yield.* 53% (based on Fe).

Elemental analysis (%) calculated (found) for $C_{30}H_{44}B_2N_{12}Fe$: C, 55.42 (55.27); H, 6.82 (6.80); N, 25.85 (26.05)%. HR-ESI$^+$: calculated for $[Fe^{II}((3, 5-(CH_3)_2Pz)_3 BH)_2]^{2+}$ 650.3352, found 650.3375; $[Fe^{II}((3, 5-(CH_3)_2Pz)_3BH)_2]^{2+}$ + Na 673.3241, found 673.3250.

The X-ray diffractogram pattern was recorded at 289 K on polycrystalline powder deposited on aluminium plate, using a Philipps Panalytical X'Pert Pro MPD powder diffractomer at Cu-kα radiation equipped with a fast detector within the 6–35° 2θ range. Supplementary Fig. 6 presents the good agreement between the powder X-ray diffractogram of the synthesized crystals and the spectra calculated from the 289 K X-ray structure of **1** (ref. 24).

**Sample preparation and STM experiments.** The Au(111) on mica samples are prepared under ultra-high vacuum by sputtering and annealing cycles (Ar$^+$ ions at 900 eV, $T = 320$ °C, base pressure: $10^{-10}$ mbar). The molecule **1** is sublimated from a home-made evaporator (Knudsen cell type) at 85 °C on the sample kept at 4.6 K. The sample is then annealed at room temperature for 30 min, to enable the formation of the molecular islands. All the experiments were realized on a Omicron STM at low temperature (4.6 K) under ultra-high vacuum (base pressure: $10^{-11}$ mbar). The STM images are acquired either in the constant current mode (topography) or in the constant height mode. The light illumination of the tunnel junction is performed using a blue laser diode (405 nm, 55 μW, 500 × 500 μm$^2$) or a He–Ne laser (632.8 nm, 55 μW, 500 × 500 μm$^2$). During light illumination, the temperature of the sample is maintained below 5 K.

**X-ray photoelectron spectroscopy.** The X-ray photoelectron spectroscopy study of a monolayer of **1** deposited on Au(111)/Mica shows that the N/Fe atomic ratio is equal to 11.5, which is in good agreement with the expected value (12) for the compound **1**. This demonstrates that the molecules perfectly survive the sublimation process and keep their integrity on the substrate (see Supplementary Fig. 7).

**Computational details.** Density functional calculations were performed using plane waves electronic structure package Quantum-ESPRESSO[31]. The generalized gradient approximation in the Perdew–Burke–Ernzerhof parametrization was employed for exchange-correlation functionals. For isolated molecules, atomic relaxations were performed for both spin states minimizing the total energy. As the HS state was found to be higher in energy (by ∼1.2 eV at equilibrium with $U = 0$), a constraint calculation was carried out in this case by fixing the total molecule spin moment.

**Data availability.** All relevant data are available from the authors on request.

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

## Acknowledgements

We thank the french national research agency ANR (ANR-BLANC-12 BS10006, SPIROU project), the labex SEAM (ANR-11-LABX-086, HEFOR project) and the Region Ile-de-France in the framework of DIM Nano-K (METEOR project) for support. The calculations were performed using HPC computation resources from GENCI-[TGCC], grant number 2015097416.

## Author contributions

O.I., M.-L.B. and T.M. designed, synthesized and characterized the molecule. F.M. performed the X-ray photoelectron spectroscopy experiment. A.K., D.L., Y.J.D., A.S. and C.B. performed and analysed the *ab initio* calculations. K.B., A.B., J.L., C.C.,Y.G., S.R. and V.R. performed, analysed and discussed the STM experiments. K.B., A.B., V.R., M.-L.B. and A.S. co-wrote the paper. All the authors discussed the results and commented on the manuscript.
