## [Peer review file · Nature Communications]

Reviewers' comments:

Reviewer #1 (Remarks to the Author):

This paper explores the properties of a particular SCO 2D molecular system on a Gold surface. The results are interesting, in that they demonstrate that by combining STM measurements with first principles calculations one is able to discriminate between the two spin states of the molecule and gain knowledge of their geometrical arrangement on the surface. These kind of studies are hardly new, and the authors acknowledge many of them in their extensive references list. Although the paper is by any means of great scientific quality and the interest in these kind of systems is high, I believe that the level of novelty in the results is not sufficient to warrant publication in Nature Communications, I would recommend the authors to seek publication on a more specialized journal.

Reviewer #2 (Remarks to the Author):

Referee report on manuscript "Molecular scale dynamics of light-induced spin crossover in a two-dimensional layer" by Kaushik Bairagi et al.

The article presents results about the spin transition of FeII spin crossover molecules deposited on Au(111). The authors demonstrate convincingly that they are able to distinguish with their STM molecules in the HS state from molecules in the LS state. This allows to show that when molecules are deposited with a sub-monolayer coverage, they form self-assembled islands of one monolayer height with a spin-dependent superstructure. In a next step, the authors studied the dynamics of spin transition upon illumination at 405 nm of this spin-dependent-superstructure. They were able to characterize the spin transition dynamics at a "macroscopic" level (scale of one image, ~50X50 nm), but thanks to the STM spatial resolution, they were also able to provide information about the internal dynamics of the spin transition.

To my knowledge, this was never done before at this scale. The manuscript is well written and the results well established. They provide interesting results about the role of cooperative effects in the spin transition of spin-crossover molecules on surfaces.

The authors may just clarify one point :

In the text they write "The distribution of tLS and tHS shown in Figure 4c are measured over more than 100 molecules", but according to fig 4c they observed 300 events during the acquisition of the first image. Maybe they can say more about the way these events were acquired, either in the main text, or in the SI.

In conclusion, I think the works deserves to be published in nat. commun.

Reviewer #3 (Remarks to the Author):

The authors have studied, combining low temperature scanning tunneling microscopy and ab initio theoretical approach based on DFT, the monolayer of spin crossover

molecule deposited on Au single crystal surface. They show that this monolayer consists in an ordered superstructure, alternating sequences of molecules in the high spin (HS) and low spin (LS) states. In addition, they evidence the transformation of this ordered overlayer upon light irradiation, as some of the low spin molecules are switched to the high spin state. The experimental data have extremely good quality, the results obtained are new, important and relevant for nanoscience and nanotechnology as they suggest routes for the manipulation of spins, down to the molecular level.

The weak point of the paper in my opinion is related to the interconnection between theory and experiments, and this point is very important in order to identify from STM data the LS and HS molecules. Modelization is performed with the DFT using GGA+U approximation. The paper does not cite other theoretical studies using similar methods for similar materials (for instance see *Phys. Chem. Chem. Phys.*, 2015,17, 16306-16314; *Magnetochemistry* 2016, 2(1), 14; *Phys. Rev. B* 78, 024433 (2008) and refs in these papers). However a short discussion of these papers would allow for instance to discuss the relevance of the present model, the relevance of calculated energetic parameters (e.g. stability of LS vs HS), the validity of calculated HOMO-LUMO, the impact of the choice of U parameter etc... Most calculations performed in the paper are performed for isolated molecules. It can be justified as a first approximation if there is a weak coupling between the molecule and the substrate (and additionally neglecting intermolecule interaction). The last paragraph of SI however presents calculations for LS molecules on Au(111). With respect to the previous remark, these data should be analyzed more in order to confirm or infirm this hypothesis (is the difference with the DOS of the free molecule representative of chemisorption, what is the adsorption energy?). In addition, comparing Fig 1c with Fig S6b, it seems that upon adsorption, $d(x^2-y^2)$ - dx_y HOMO states move close to the Fermi energy and LUMO states are shifted by $\sim 2\text{eV}$, suggesting that adsorption has a direct effect on the molecular levels and that the distinction between LS and HS states from electronic effects needs more careful discussion. In that sense I would suggest to present (eg in SI) comparison between experimental dI/dV and theoretical DOS in a more extended range ($\sim \pm 1\text{eV}$), so that spectral regimes dominated by electronic transitions or vibrational modes show up more clearly. Additionally, I do not understand why calculations cannot be performed for HS molecules on Au(111), as previously done by other groups on similar systems (see *Phys. Rev. B* 87, 144413 (2013)), this would allow comparison of experiments with a more realistic theory.

These points have to be discussed convincingly before the paper can be accepted.

Authors' response to reviewer comments

We thank the three reviewers for their very positive comments on our work and their constructive remarks. We provide detailed answers to all their comments in the following:

Reviewer #1:

We thank the reviewer 1 for his/her comments. He/she recognized the 'great scientific quality' of our work and the high interest in spin-crossover molecular systems but is not convinced by the level of novelty of our work, in contrary to reviewers 2 and 3 who both underlined the novelty and importance of our results.

We would like to emphasize in the following why we believe that our work is indeed new and deserves publication in Nature Communications.

Firstly, we have been able to identify unambiguously the spin state of single spin-crossover molecules by inelastic tunneling spectroscopy in combination with ab initio calculations, for the first time to the best of our knowledge. This is new and should be a more versatile signature than the zero bias Kondo feature observed so far [9,13] that needs a priori a rather strong coupling with a metallic surface and sufficiently low temperature (below the Kondo temperature).

Secondly, we have observed by scanning tunneling microscopy (STM) a new thermodynamics phase of FeII pyrazolylborate molecules in low dimension (ordered mixed phase of high spin and low spin molecules) that does not exist in the bulk. We believe that this observation is certainly more general to other spin-crossover molecules and can explain recent results of incomplete phase transformations observed by x-ray absorption (fig. 5 of ref. 12 for example).

Last but not least, we have measured for the first time by STM the light induced excited spin state trapping (LIESST) and its molecular scale dynamics. To the best of our knowledge, this effect has been measured at a monolayer level only by a x-ray absorption technique [12]. It is worth noting that the use of x-ray is well known in the community to be very powerful but also to have severe drawbacks, namely the degradation of molecules under the x-ray beam and the soft x-ray induced excited spin state trapping (SOXIESST) that generally make the analysis of the results rather complex. STM is a low energy probe that is known to preserve the molecules and their spin state and the analysis of the LIESST effect is therefore more direct. Moreover, it allows an unprecedented molecular resolution on the dynamics of the LIESST, showing cooperative effects down to the molecular scale for the first time, what is of importance for the large community of chemists working since a long time on the dynamics in the bulk phases.

To better emphasize those novelties, we have modified some sentences of the introduction.

p.1 c.1 l.28, we have added 'While x-ray absorption techniques can provide valuable information, they remain macroscopic and present important drawbacks like the degradation of molecules and soft x-ray induced spin state trapping. Therefore, scanning tunneling microscopy (STM) stands as a unique technique to access the direct visualization of photo-induced spin transition at the molecular scale'.

p.1 c.2 l.23, we have added 'This local inelastic spectroscopy should provide a more versatile way of determining the spin state of spin-crossover molecules than the zero bias Kondo feature observed so far that required a strong coupling with a metallic substrate and low temperature measurements.'

p.1 c.2 l.28, we have added 'We thus provide evidences of the formation of a long-range order superstructure alternating the presence of one molecule in HS state and two molecules in the LS state, which is a new thermodynamics phase of those spin-crossover molecules in low dimension as compared to the bulk.'

p.2 c.1 l.3, we have added 'STM measurements enable us to access for the first time the molecular scale dynamics of the spin-state switching...'

Reviewer #2:

We thank the reviewer 2 for his/her positive comments on our work and to have underlined the novelty of our results. The reviewer asks a question on fig. 4c and would appreciate a more precise description on the acquisition of time profile of fig. 4b. The reviewer wonders why we measure 300 events at short time whereas we have measured time profiles over 100 molecules. This confusion comes from our text which is not clear enough on the definition of t_{LS} and t_{HS} which is not a time but a duration. This means that short t_{LS} does not correspond to the first image but to events where the molecules switch between two consecutive images. It is therefore clearer why we can measure 300 events at short time out of 100 studied molecules: the same molecule can switch back and forth, as shown in fig. 4b and therefore contribute several time to the statistics of the duration between two switching events. We have clarified this in the text by replacing 'time' by 'duration' and have better explained how these events were acquired.

p.5 caption of Fig. 4, we have added 'Distribution of t_{LS} and t_{HS} , respectively the duration for a molecule in a LS (HS) state before switching in a HS (LS) state.'

p.5 c.2 l.13, we have added 'Technically, this has been done by subtracting a mean base plane to all raw STM images and measuring the mean height over a small region of interest that stays focused on the very same molecule for more than hundred images.'

Reviewer #3:

We thank the reviewer for his/her very positive general comment on our work. We also thank him/her for his/her critical reading of the manuscript that greatly helped to improve the theoretical part and its comparison with experiments. In the following, we answer point by point to all the remarks:

-we have better discussed the relevance of our present model and its relevant parameters, citing the four papers suggested by the reviewer that were indeed particularly suited. We have added three paragraphs, one in the introduction, on when discussing the energetic and the effect of U and one when presenting the results in comparison with experiments to better explain the context.

p.1 c.2 I.2, we have added the following paragraph: 'From theoretical point of view, SCO molecules have been studied in several works within so-called DFT+U approach which combines the Density Functional Theory with the Hubbard U onsite term (applied on localized orbitals of magnetic atom), necessary to correct for self-interaction errors. For example, Lebegue *et al.* \cite{Lebegue2008} and Paulsen *et al.* \cite{Paulsen2016} have used GGA+U to study molecular crystals of Fe-based SCO molecules. Also, DFT studies of SCO molecules on metallic substrates \cite{Gueddida2013} or bidimensional materials \cite{Garcia2015} have been recently presented. As a result, the importance of the U parameter for describing properly the stability and LS to HS transition was pointed out.'

p.2 c.1 I.10, we have added 'In bulk, spin crossover molecules \cite{1} (Figure~\ref{Fig_STM})a) present a transition from LS to HS at a temperature of ca. 186~K or by LIESST effect \cite{Davesne2015}. Our theoretical DFT+U study confirms that the molecular magnetism is tightly related to the Fe-N distance -- the longer it is the more favorable is the HS solution (see SI, Figure~S4) -- a well-known fact resulting from the competition between the Hund's rule coupling and the Fe d -levels splitting in a crystal field. The inclusion of U on the Fe d -orbitals does not change significantly HS and LS atomic configurations but reduces significantly the HS-LS energy separation, from \$1.2\$ (U=0) to \$0.6\$ (U=2 eV) (Figure~S4), favoring further the HS state.'

p.2 c.2 I.41, we have added 'In order to assign the spin state of the bright and dark molecules at 0.3~V and understand their round shape observed by STM, we performed *ab initio* density-functional calculations following a well established DFT+U procedure. As already discussed above, the magnetism is favored by U, but the main physical parameter underlying the magnetic transition is the Fe-N distance. Since the exact value of U remains unknown (usually it is in the range between 1.5 and 3 eV for Fe) we will present and discuss in the following the results of DFT calculation without U in order to get physical interpretation of experiments. We have checked however that the major influence of U on the electronic structure is an increase of a HOMO-LUMO gap while the main physics at the orbital level remains unchanged (see SI, Figure~S7).'

- we thank the reviewer for noting the disagreement between relative level positions in Fig. 1 and Fig. S6. In fact the PDOS shown in Fig. 1 was calculated for a non relaxed molecule that could not compare well with the one on the substrate that was relaxed. We have now replaced Fig. 1 with the correct data calculated at the equilibrium configurations for both spin states. Since for isolated molecule the Fermi level is not well defined (it could be in any place between HOMO and LUMO) we have put as the zero of energy the positions of HOMOs instead of an arbitrary position between HOMO and LUMO as done in the previous version. For these correct geometries we indeed observe that the levels of a free molecule are only slightly modified by the substrate. We have added some discussion on this and also the value for adsorption energy confirming weak molecule-substrate interaction.

p.3 c.1 I.10, we have added 'This important point is confirmed for the LS molecule deposited on the Au(111) surface (see SI, Figure~S6) where only $d_{x^2-y^2,xy}$ -orbitals are found to be seen in the vacuum above the molecule while no signal is observed for the d_{z^2} -originated states (we did not calculate the deposited molecule in the HS state since it needs a locally constraint magnetic calculation, otherwise it converges to the lowest energy LS state, which is not yet implemented in our code). It has been also found that molecular levels and their relative positions are

only weakly modified upon adsorption on the Au surface (the adsorption energy was found to be about \$2.1\$ eV which also indicates a rather weak molecule-substrate interaction) justifying thus our free molecule analysis.'

- the calculation of magnetic state for the deposited molecule is quite delicate since it is much higher in energy with respect to the LS state, so if one just starts from initial magnetic guess one will converge to the final lowest energy nonmagnetic solution during the self-consistent run. This does not probably happen for the molecule studied in Gueddida2013, may be due to smaller HS-LS energy separation so that the solution remains magnetic *throughout* the self-consistency cycle. For the free molecule we could do a constraint calculation fixing the total spin moment. But this procedure did not work well if the molecule is deposited on nonmagnetic Au surface since the spin moment gets spread over the whole molecule+Au system during the self-consistency, which is unphysical. Anyway, the calculation of deposited molecule was done just to confirm our two key points: i) molecular levels are not much modified upon adsorption (weak molecule-substrate interaction) ii) only $d_{\{xy,x^2-y^2\}}$ is important for the tunneling transport measured by STM. We have added some phrases about this point in the paragraph discussed in the previous question and in the SI.

Reviewers' Comments:

Reviewer #2 (Remarks to the Author)

Referee report on manuscript "Molecular scale dynamics of light-induced spin crossover in a two-dimensional layer" by Kaushik Bairagi et al.

The authors modified the manuscript to take into account the comments made by the referees. In particular, they better explain how the data shown in fig. 4c were acquired. The quality of the manuscript has improved, and I still think the work deserves to be published in nat. commun.

Reviewer #3 (Remarks to the Author)

The authors replied convincingly to my criticisms. Taking into account the excellent quality of the data and the importance of the results for the community of researchers in physics and chemical physics, I strongly recommend the manuscript from Bairagi et al. for publication in Nature Communications.